# Locating and Editing Factual Associations in Mamba

**Arnab Sen Sharma,** * **David Atkinson, and David Bau**
Khoury College of Computer Sciences, Northeastern University

## Abstract

We investigate the mechanisms of factual recall in the Mamba state space model. Our work is inspired by previous findings in autoregressive transformer language models suggesting that their knowledge recall is localized to particular modules at specific token locations; we therefore ask whether factual recall in Mamba can be similarly localized. To investigate this, we conduct four lines of experiments on Mamba. First, we apply causal tracing or interchange interventions to localize key components inside Mamba that are responsible for recalling facts, revealing that specific components within middle layers show strong causal effects at the last token of the subject, while the causal effect of intervening on later layers is most pronounced at the last token of the prompt, matching previous findings on autoregressive transformers. Second, we show that rank-one model editing methods can successfully insert facts at specific locations, again resembling findings on transformer LMs. Finally we adapt attention-knockout techniques to Mamba in order to dissect information flow during factual recall. We compare Mamba directly to a similar-sized autoregressive transformer LM and conclude that despite significant differences in architectures, when it comes to factual recall, the two architectures share many similarities.

## 1 Introduction

Studies of autoregressive transformer language models' (LMs) processing of factual statements such as *The Eiffel Tower is located in Paris* have identified a localized pattern of internal computations when recalling facts (Meng et al., 2022a;b; Geva et al., 2023; Hernandez et al., 2023; Nanda et al., 2023), and have further found that those LMs can be edited by making single-layer rank-one changes in model parameters to alter a specific fact. Although these localized phenomena appear to generalize across autoregressive transformer LMs, the extent to which similar locality might appear in very different architectures—such as recurrent networks (RNNs)—has not yet been investigated.

In this paper we investigate the internal mechanisms of Mamba (Gu & Dao, 2023), a recently-proposed state-space language model, a type of RNN that achieves per-parameter performance that is competitive with transformers. Specifically, we ask whether factual recall within Mamba exhibits locality similar to the patterns observed in autoregressive transformer language models.

Our paper is a case study confronting a key methodological challenge that broadly faces interpretability researchers: as state-of-the-art neural network architectures evolve, we must ask, can the detailed analytical methods and tools developed for one neural architecture, such as transformer LMs, be generalized and applied to a different neural architecture, such as Mamba? In this paper we are able to answer the question with a qualified "yes": we find that many of the methods used to analyze transformers can also provide insights on Mamba. We also discuss mismatches—that is, interpretation methods (such as attention-knockout) that do not transfer to Mamba as easily due to architectural constraints.

We begin by studying whether activation patching (Wang et al., 2022) can be successfully applied to Mamba. Known variously as causal mediation analysis (Vig et al., 2020), causal

---

*Correspondence to sensharma.a@northeastern.edu, Code available at `romba.baulab.info`

tracing (Meng et al., 2022a), and interchange interventions (Geiger et al., 2021), activation patching techniques can successfully identify specific model components in transformer LMs that play crucial roles in performing a task. We ask whether Mamba can be productively studied the same way, even though the architectural components of Mamba are very different: for example, instead of attention heads and MLP modules, Mamba is composed of convolutions, gates, and state-space modules. To answer, we adapt activation patching to Mamba, and ask if any sparsity patterns emerge which provide insights into the respective roles of its components.

We also study whether rank-one model editing can be applied to Mamba. While studies of transformers (Meng et al., 2022a;b; Hase et al., 2024) have found that there are a range of MLP modules within which factual knowledge can be inserted by making a single rank-one change in parameters, Mamba does not have MLP modules, so we ask if there are any other modules that can be similarly edited to insert knowledge. As with previous studies of transformers, the key question is whether factual associations can be edited with both specificity (without interfering with unrelated facts) and generalization (while remaining robust to rewordings of the edited fact).

Finally, we apply methods for understanding the overall information flows in Mamba. Inspired by Geva et al. (2023), we adapt attention-blocking methods to the attention-free Mamba LMs.

In this work we conduct our experiments on Mamba-2.8b, the largest available LM in Mamba family, and for comparison we conduct the same experiments on the similarly sized Pythia-2.8b (Biderman et al., 2023) autoregressive transformer LM.

## 2 Background on Mamba

Mamba, introduced in Gu & Dao (2023), is a recent family of language models based on state space models (SSMs). SSMs are designed to model the evolution of a hidden state across time with a first-order differential equation (Koopman et al., 1999; Durbin & Koopman, 2012); when they are used as the recurrent state of an RNN, they can enable highly efficient parallelized training (Gu et al., 2021). To achieve good performance in language modeling, the Mamba SSM introduces input-dependent parameterization or *selective*-SSM instead of the traditional time-invariant SSMs. Mamba uses a special architecture called **MambaBlock**[1], which is stacked homogeneously, replacing both attention and MLP blocks used in transformer layers. Here, we focus on the different operations performed inside a MambaBlock.

Formally, Mamba is an autoregressive language model: $M : \mathcal{X} \to \mathcal{Y}$ over a vocabulary $\mathcal{V}$ that maps a sequence of tokens $x = [x_1, x_2, \ldots, x_T] \in \mathcal{X}$, $x_i \in \mathcal{V}$ to $y \in \mathcal{Y} \subset \mathbb{R}^{|\mathcal{V}|}$ which is a probability distribution over the next token continuations of $x$. Similar to other deep LMs, in Mamba, a token $x_i$ is first embedded to a hidden state of size $d$ as $h_i^{(0)} = emb(x_i)$. Then $h_i^{(0)}$ is transformed sequentially by a series of MambaBlocks. The hidden state $h_i^{(\ell)}$ after the $\ell^{th}$ (1-indexed) MambaBlock is computed as follows:

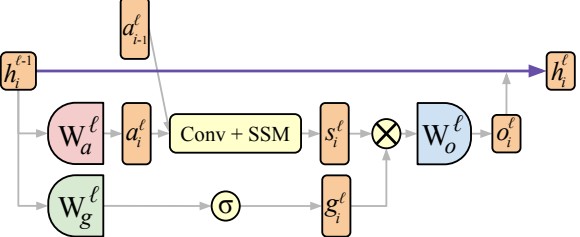

Figure 1: Architecture of a MambaBlock. Projection matrices $W_a^\ell$ and $W_g^\ell$ have the shape $2d \times d$, while $W_o^\ell$ has the shape $d \times 2d$. $h, a, g, s,$ and $o$ are intermediate states of a token representation. $\sigma$ is SiLU activation and $\otimes$ is elementwise multiplication. *Conv + SSM* operation abstracts the Conv1D and *selective*-SSM operations.

$$h_i^{(\ell)} = h_i^{(\ell-1)} + o_i^{(\ell)} \tag{1}$$

where $o_i^{(\ell)}$ is the output of $\ell^{th}$ MambaBlock for the $i^{th}$ token

---

[1]In their paper, Gu & Dao (2023) call this component Mamba—the same name as the LM family.

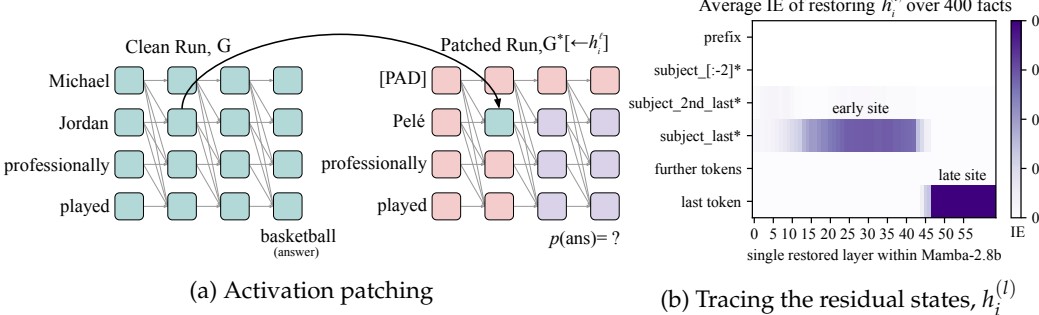

(a) Activation patching

(b) Tracing the residual states, $h_i^{(l)}$

Figure 2: **(a)** Activation patching. A state from the clean run $G$ is *patched* into its corresponding position in the corrupted run $G^*$. This has a downstream effect of potentially changing all the states that depend on the patched state in $G^*[\leftarrow h_i^{(\ell)}]$. **(b)** Average IE of applying causal tracing on residual stream states ($h_i^{(\ell)}$ in Figure 1) across 400 different facts from the RELATIONS dataset (See Appendix A.2).

$$o_i^{(\ell)} = \text{MambaBlock}^{(\ell)}\left(h_1^{(\ell-1)}, h_2^{(\ell-1)}, \ldots, h_i^{(\ell-1)}\right) = W_o^{(\ell)}\left(s_i^{(\ell)} \otimes g_i^{(\ell)}\right) \quad (2)$$

Here, $\otimes$ represents element-wise multiplication or Hadamard product. $s_i^{(\ell)}$ is calculated as:

$$a_i^{(\ell)} = W_a^{(\ell)} h_i^{(\ell)} \quad (3)$$

$$c_1^{(\ell)}, c_2^{(\ell)}, \ldots, c_i^{(\ell)} = \text{SiLU}\left(\text{Conv1D}\left(a_1^{(\ell)}, a_2^{(\ell)}, \ldots, a_i^{(\ell)}\right)\right) \quad (4)$$

$$s_i^{(\ell)} = \textit{selective-}\text{SSM}\left(c_1^{(\ell)}, c_2^{(\ell)}, \ldots, c_i^{(\ell)}\right) \quad (5)$$

We abstract the operations in Equations 4 and 5 as the **Conv + SSM** operation in Figure 1. At a high level, Conv + SSM brings information from the past token representations to the current token representation, similar to the *attention* blocks in transformer LMs. But, unlike attention operation, Conv + SSM scales *linearly* with the context length and thereby enjoys faster inference speed and longer context limits. See Gu & Dao (2023) for details.

The output of the other path $g_i^{(\ell)}$ (that does not pass through Conv + SSM operation) is a gating mechanism that regulates the information flow. This gating mechanism resembles parts of LSTM (Hochreiter & Schmidhuber, 1997) and GRU (Cho et al., 2014) networks, where similar gates control selective updates of recurrent state.

$$g_i^{(\ell)} = \text{SiLU}\left(W_g^{(\ell)} h_i^{(\ell-1)}\right) \quad (6)$$

In the remainder of the paper, we aim to characterize the role of the components of Mamba in factual recall by adapting tools that have previously been used to analyze transformers. In Section 3, we apply activation patching to localize factual recall as in Meng et al. (2022a), testing the roles of states $s_i$, $g_i$, and $o_i$ at all layers. In Section 4, following Meng et al. (2022a); Hase et al. (2024), we test rank-one edits of facts across components $W_a$, $W_g$, and $W_o$ at each layer. And in Section 5 we address the challenge of applying attention patching in Mamba, as used in Geva et al. (2023) to isolate information flow in GPT LMs. We include a further investigation of Mamba's factual recall mechanisms in Appendix E, following Hernandez et al. (2023) in collecting Jacobians within Mamba to test the linearity of its relational encodings.

## 3 Locating Key States for Factual Recall

We begin with activation patching, seeking to understand if there are specific hidden states which play important roles during factual recall. We select a fact $(s, r, o)$ that the LM knows, where $r$ is a relation that associates a subject entity $s$ with an object entity $o$. To estimate each state's contribution towards a correct factual prediction ($s = \textit{Michael Jordan}$, $r = \textit{professionally played}$, $o = \textit{basketball}$), we collect model activations across three different runs:

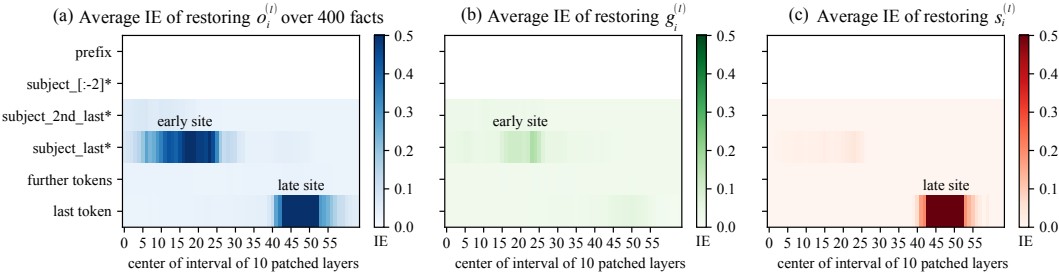

Figure 3: Average indirect effect of different states $o_i^{(\ell)}$, $g_i^{(\ell)}$, and $s_i^{(\ell)}$ over 400 facts from the RELA-TIONS dataset (see Appendix A.2). For each layer $\ell$, states for a window of 10 layers around $\ell$ are restored from the clean run $G$.

**clean run** $G$: In the clean run, we run the model on a prompt specifying the fact we are interested in. For example, $x = (s, r) = $ *Michael Jordan professionally played*. We cache all the hidden states during the clean run to be used later: $\left\{ h_i^{(\ell)}, a_i^{(\ell)}, s_i^{(\ell)}, g_i^{(\ell)} \mid i \in [1, T], \ \ell \in [1, L] \right\}$.

**corrupted run** $G^*$: In the corrupted run, we swap $s$ with a different subject $s^*$ (*Pelé*) such that the LM gives a different answer $o^*$ (*soccer*) to the modified prompt $x^* = (s^*, r)$ (i.e., $o^* \neq o$).

This subject-swapping approach follows the recommendation of Zhang & Nanda (2023) and has the advantage of using natural text perturbations to avoid introducing out-of-domain states to the model's computation, as may happen when corrupting $s$ embeddings with Gaussian noise (the method used in Meng et al. (2022a)).

**patched run** $G^*[\leftarrow h_i^{(\ell)}]$: In the patched run, we run the model on the corrupted prompt $x^*$, but intervene on $h_i^{(\ell)}$ by replacing its value with the corresponding state cached from the clean run $G$. The remainder of the computation is run normally, meaning that the patched state can have a downstream effect of potentially changing all the states that depend on it. See Figure 2a.

Let $p(o)$, $p^*(o)$, and $p^*[\leftarrow h_i^{(\ell)}](o)$ denote the probability assigned to the correct answer $o$ in $G$, $G^*$, and $G^*[\leftarrow h_i^{(\ell)}]$ respectively. To measure the contribution of $h_i^{(\ell)}$ in recalling the fact $(s, r, o)$, we define its *indirect effect* (IE) as:

$$\text{IE}_{h_i^{(\ell)}} = \frac{p^*[\leftarrow h_i^{(\ell)}](o) - p^*(o)}{p(o) - p^*(o)} \tag{7}$$

In Figure 2b we plot the average indirect effect of restoring the residual states $h_i^{(\ell)}$ across different layer-token positions over 400 facts from the RELATIONS dataset (Hernandez et al., 2023). The high IE observed at the *late site* (later layers at the last token) position is natural, as restoring a clean $h_i^{(\ell)}$ there will restore most of the model computation from $G$. However, Mamba also shows high causality at the *early site* (early-middle layers at the last subject token position). This is consistent with what Meng et al. (2022a) observed in GPT LMs.

In Figure 3 we plot the average IE for $o_i^{(\ell)}$, $g_i^{(\ell)}$, and $s_i^{(\ell)}$. The plot for $o_i^{(\ell)}$ (Figure 3a) looks very similar to Figure 2b, confirming that the output from MambaBlock has strong causal effects at both early and late sites. Interestingly, Figure 3c shows that the selective-SSM outputs $s_i^{(\ell)}$ have high IE only at the late site, resembling the behavior of attention modules in GPT models (Meng et al., 2022a). However, there is no state that appears to do the opposite; in other words, there is no state with strong effects at the early site and not at the late site (The gate output $g_i^{(\ell)}$ does have stronger IE at the early site, but these effects are very weak). To compare with autoregressive transformer LMs, activation patching results for Pythia-2.8b is shown in Figure 9 in Appendix D. This comparison reveals a key way how Mamba differs from transformers: while transformer MLP outputs have effects in the early

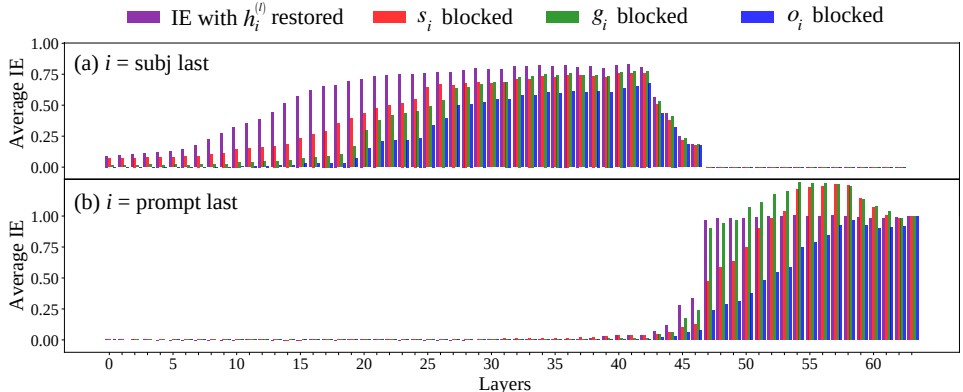

Figure 5: Impact of ablating $s_i$, $g_i$, and $o_i$ on $\text{IE}_{h_i^{(\ell)}}$ for **(a)** *subject last* and **(b)** *prompt last* token positions. Taken together (a) and (b) show a clear separation roles between early-mid and later layers in Mamba-2.8b. $h_i^{(\ell)}$ up to layer 46 only show strong IE at the subject last token position and have negligible impact after that. Whereas IE of $h_i^{(\ell)}$ jumps to 1.0 after layer 46. **(a)** also shows that, at the subject last token, before layer $27 - 28$, $\text{IE}_{h_i^{(\ell)}}$ is significantly reduced by blocking either $o_i$, $g_i$, or $s_i$ paths (sorted in descending order of damaging $\text{IE}_{h_i^{(\ell)}}$). **(b)** At the prompt last token, ablating $o_i$ or $s_i$ paths can significantly reduce $\text{IE}_{h_i^{(\ell)}}$ in layers $47 - 50$.

site and not the late site, in Mamba there is no similar state that specializes only at the early site, at which factual recall would be expected to occur. This presents the question: which parameters in Mamba mediate factual recall?

To investigate this question, we replicate an experiment from Meng et al. (2022a) to probe *path-specific effects* (Pearl, 2022) by severing a path from the causal graph and monitoring its effect. Here, we are interested in understanding the effect of the contributions from $g_i$, $s_i$, and $o_i$ (i.e. states that are processed by $W_g$, Conv + SSM, and $W_o$ respectively) while recalling a fact. First, in the corrupted run $G^*$, at token position $i$, we cache all the contributions from the $s_i$ paths as $s_i^* = \{s_i^{*(\ell)} | \ell \in [1, L]\}$. Then in the patched run $G^*[\leftarrow h_i^{(\ell)}]$, we restore $h_i^{(\ell)}$ that was cached from the clean run $G$ into its corresponding state (as in Figure 2a), but with an additional modification: to understand the contribution from the $s_i$ paths, we sever those paths by also patching $s_i^*$ (cached from the corrupted run $G^*$) to their

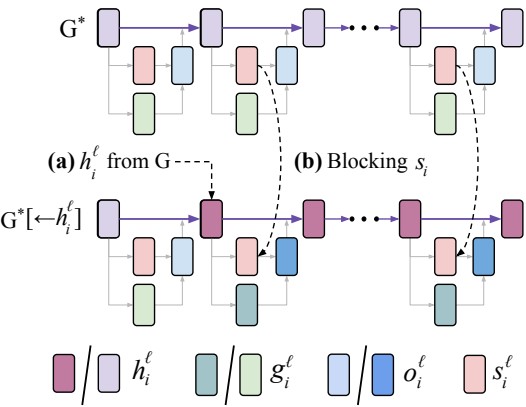

Figure 4: To probe for path-specific effects, **(a)** $h_i^{(\ell)}$ is restored from the clean run $G$ (as in Figure 2a). **(b)** Then, to reveal the role of the Conv + SSM contributions, $s_i^*$ states from the corrupted run $G^*$ are also patched to block the contributions from $s_i$ paths.

corresponding locations (see Figure 4). The same experiment is replicated to understand the contributions of $g_i$ and $o_i$ states. We note that severing the $o_i^{(\ell)}$ will sever $s_i^{(\ell)}$ and $g_i^{(\ell)}$ as well (see Figure 1).

In Figure 5 we plot the average results of this experiment for token positions (a) $i =$ *subject last* and (b) $i =$ *prompt last* over 400 examples randomly sampled from the RELATIONS dataset. The key findings can be understood by examining the gap between the purple bars and the red, green, and blue bars: a large gap indicates a strong mediating role for Conv + SSM, $W_g$, or $W_o$ parameters, respectively. At the early site at the subject last token, both the Conv + SSM and $W_g$ have a strong role, but $W_o$ plays an even larger role than either. Yet the strongest mediator at the late site is also $W_o$. This experiment highlights the importance of $W_o$ in both stages of predicting a fact. But it also suggests that Mamba

does not separate early-site factual recall between these groups of parameters as cleanly as transformers. However, Figure 5 reveals a clean separation of roles between early to mid and later layers, analogous to the findings of Hernandez et al. (2023) in transformer LMs. We also note that this division of responsibilities between layers can be more sharply noticed in Mamba when compared to transformers LMs (compare Figure 5 with Figure 10).

## 4   Editing Facts With ROME

Having begun to characterize the locations of important states for factual recall, we now investigate whether factual recall behavior can be edited. In particular, we apply the ROME (Rank One Model Editing, Meng et al., 2022a) technique to Mamba. ROME begins with the observation that any linear transformation can be considered as an associative memory (Anderson, 1972; Kohonen, 1972), mapping a set of keys $K = [k_1|k_2|\dots]$ to their corresponding values $V = [v_1|v_2|\dots]$, and uses this to edit factual associations in transformer LMs. Here, we apply the technique to a particular set of linear transformations within Mamba, and report our editing success on each.[2]

The input to ROME is a prompt $x = (s, r)$, where $s$ (*Emmanuel Macron*) is a subject entity and $r$ (*is the President of*) is a relation. ROME also takes a counterfactual object $o^*$ (*England*), meant to replace the correct object $o$ (*France*) in the model's output. To effect that change, ROME generates a rank-one update to $W_{down}^{(\ell)}$, the down-projection matrix of the MLP module for the last token of the subject at layer $\ell$—which plays the role of the associative memory. In generating the rank-one update, ROME considers the input to $W_{down}^{(\ell)}$ as the *key* ($k_*$). Then, with gradient descent ROME calculates a *value* ($v_*$) such that, when $v_*$ is inserted as the output of $W_{down}^{(\ell)}$, the model will output $o^*$. Importantly, while optimizing $v_*$, ROME attempts to minimize unrelated changes in model outputs (*Joe Biden*, for example, should still be mapped to *the United States* post-edit). Finally, ROME adds a rank-1 matrix $\Delta$ to $W_{down}^{(\ell)}$ such that $(W_{down}^{(\ell)} + \Delta)k_* \approx v_*$. (See Meng et al. (2022a) for details.)

### 4.1   Applying ROME in Mamba

We apply ROME on the three different projection matrices of Mamba: $W_a^{(\ell)}$ which affects only the Conv + SSM path, $W_g^{(\ell)}$ which affects only the gating path, and $W_o^{(\ell)}$, the final output of the MambaBlock, which is added to the residual state. We plot ROME performance on different projection matrices $(W_a^{(\ell)}, W_g^{(\ell)},$ and $W_o^{(\ell)})$ across all the layers in Figure 6a.

To evaluate editing performance, we use the COUNTERFACT dataset from Meng et al. (2022a). COUNTERFACT contains 20K counterfactual examples in the form $(s, r, o \to o^*)$, where $o$ is the correct answer to the prompt $x = (s, r)$, and $o^*$ is the object which is to be inserted as the new answer to the prompt (See Appendix A.1 for details). We select the first 2000 examples from this dataset for our module-layer sweep. We use the original evaluation matrices in Meng et al. (2022a) to measure ROME edit performance. The final **score (S)** in the ROME evaluation suite is the harmonic mean of three different scores:

1. **Efficacy (ES)**: For an edit request $(s, r, o \to o^*)$, we say the edit is *effective* if, post-edit, the LM assigns $p(o^*) > p(o)$ in response to the prompt $x = (s, r)$. Efficacy reflects the portion of the examples where the edit was effective.
2. **Generalization (PS)**: A successful edit should be persistent across different paraphrases of $(s, r)$. For each of the request instances $(s, r, o \to o^*)$, $p(o^*) > p(o)$ is checked post-edit with a set of different rephrasings $x_p \sim \mathcal{P}_r(s)$ of the prompt $x = (s, r)$, where $\mathcal{P}_r$ denotes a set of paraphrased templates for the relation $r$.

---

[2]Further motivating these experiments, previous work has shown that the locations identified by activation patching techniques are not necessarily those which have the strongest edit performance (Hase et al., 2024).

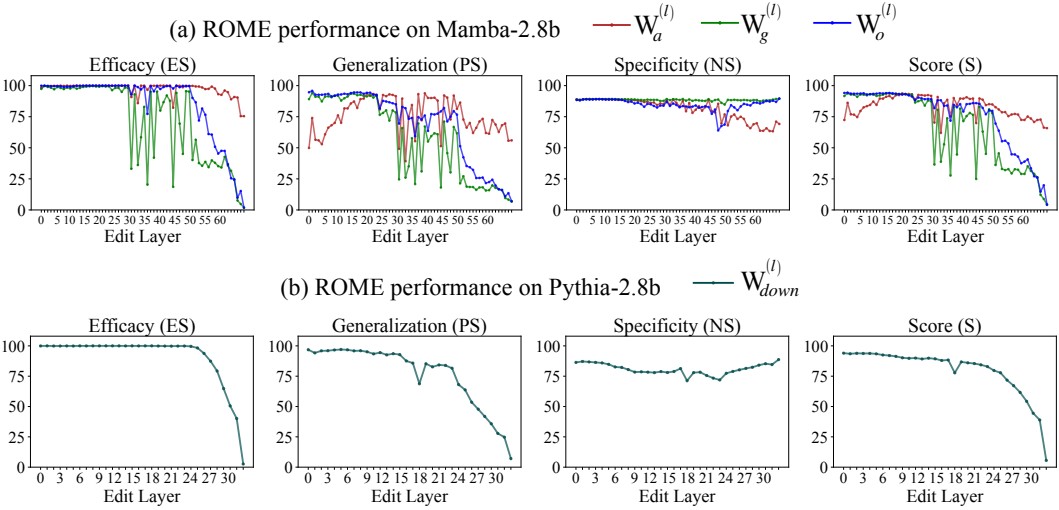

Figure 6: ROME performance in editing facts across different layers **(a)** by modifying $W_a^{(\ell)}$, $W_g^{(\ell)}$, $W_o^{(\ell)}$ in Mamba-2.8b, and **(b)** modifying $W_{down}^{(\ell)}$ in Pythia-2.8b. Results are reported on the first 2000 examples in the COUNTERFACT dataset.

3. **Specificity (NS)**: Finally, the edit should be specific to $\mathcal{P}_r(s)$ and should not additionally change the mapping of some nearby subject $s_n$ to $o^*$. To evaluate the specificity of an edit we measure $p(o_n) > p(o^*)$ with $\mathcal{P}_r(s_n)$ for a set of nearby factual associations $\{(s_n, r, o_n) \mid o_n \neq o^*\}$.

Figure 6a shows that ROME can achieve high scores (S) for a range of early to middle layers by modifying any one of the projection matrices $W_a^{(\ell)}$, $W_g^{(\ell)}$, or $W_o^{(\ell)}$, matching observations made by Hase et al. (2024) regarding transformer LMs. However, we found that performance does depend on the location of the edit. For example, in the case of $W_g^{(\ell)}$ and $W_o^{(\ell)}$, the score (S) and generalization (PS) drops after around layer 43. This is consistent with our findings from the path-blocking experiment in Figure 5a. We also find that edits to $W_a^{(\ell)}$ have poor generalization (PS) in early layers, whereas high PS can be achieved at early layers by modifying either $W_g^{(\ell)}$ or $W_o^{(\ell)}$, consistent with their higher indirect effects as seen in Figure 5a.

Where is the right place to apply ROME on Mamba? Figure 3 could suggest $W_g^{(\ell)}$, since the causal effect of $g_i$ states is mostly concentrated at the subject last token, similar to the behavior of MLPs in transformers (Meng et al., 2022a). Consistent with this is the architectural fact that, just as transformers' $W_{down}^{(\ell)}$ connects to attention modules only through the residual stream, the output of $W_g^{(\ell)}$ does not flow through the Conv + SSM module—a module that other work has suggested might play a role similar to that played by attention heads in transformers (Grazzi et al., 2024). And, indeed, we find that ROME can successfully insert facts by modifying $W_g^{(\ell)}$. On the other hand Figure 6a reveals sudden drops in efficacy and generalization at middle layer gates, suggesting that $W_g^{(\ell)}$ may be an unreliable mediator at some layers. Our experiments further show that the best performance for ROME is empirically achieved by modifying $W_o^{(\ell)}$. This is consistent with the fact that $o_i$ states show a stronger causal effect at the subject last token than $g_i$ states do (see Figures 3a and 5a). Additionally, ROME achieves better generalization (PS), competitive specificity (NS), and an overall better score (S) with $W_o^{(\ell)}$. We hypothesize that the strong performance of $W_o^{(\ell)}$ may be due to the the separation of roles between early-mid and later layers observed in Figures 2b, 3a, and 5. Also see Appendix C where we isolate the contribution of $W_o^{(\ell)}$ by

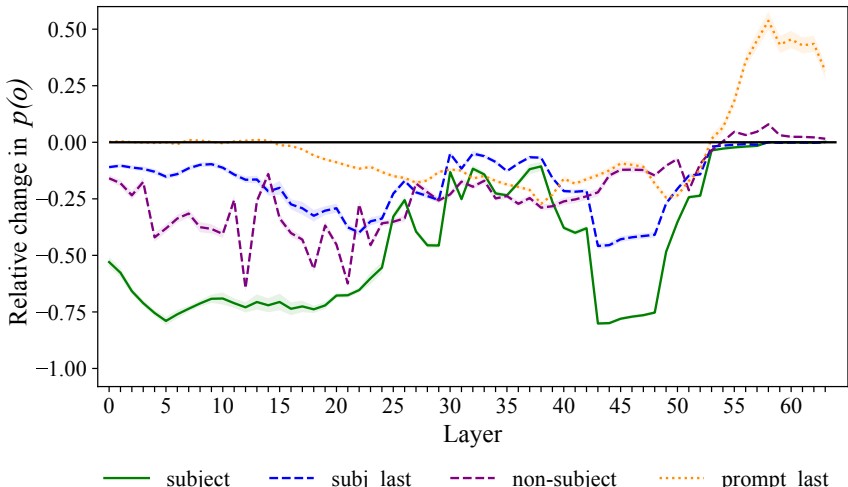

Figure 7: Relative change in $p(o)$ when information flow from $a_k^{(\ell)}$ to future tokens via $s_i$ paths is blocked, with $k$ taking the value of either *subject*, *non-subject*, or the *prompt_last* token positions. For each layer $\ell$, $s_i$ paths were blocked for a window of 10 layers around $\ell$.

subtracting $\text{IE}_{s_i^{(\ell)}} + \text{IE}_{g_i^{(\ell)}}$ from $\text{IE}_{o_i^{(\ell)}}$, which reveal a critical role of $W_o^{(\ell)}$ in early-mid layers at subject last token position while mediating a fact.

We plot ROME performance for a similar sized Pythia model on Figure 6b for comparison.

## 5   Attention Knock-out in Mamba?

Attention modules mediate the flow of information across different token positions in transformer LMs. In attention *"knock-out"* experiments the information that flows through a specific edge (from $k^{th}$ token to $q^{th}$ token) via a certain attention head is blocked to understand if critical information flows through that edge. This is also a form of causal mediation analysis and it has been effective in understanding the information flow in transformer LMs (Geva et al., 2023; Wang et al., 2022; Todd et al., 2023). In Mamba, information from past tokens is retained in the $s_i$ states, with the Conv + SSM operations (see Figure 1 and Equations 3–5). We ask, can we perform experiments similar to attention knock-out experiments in Mamba in order to understand how it moves factual information?

**We find that performing similar experiments in Mamba can be difficult**. The use of Conv with a non-linearity in conjunction with selective-SSM make it challenging to remove the information retained in the $q^{th}$ token from the $k^{th}$ token (see Appendix B for details). However, it is possible to block the propagation of information from the $k^{th}$ token to *all* the future tokens via Conv + SSM with mean-ablation. Specifically, for a layer $\ell$, we set $a_k^{(\ell)} := \mathbb{E}\left[a^{(\ell)}\right]$, where $\mathbb{E}\left[a^{(\ell)}\right]$ is the mean of $a^{(\ell)}$ states collected with 10,000 tokens from WikiText-103 by Merity et al. (2016). We recognize that this intervention may not be as *surgical* as cutting a specific edge. However, with some caveats, this experiment suggests that the factual information flow in Mamba is similar to what Geva et al. (2023) observed in GPT LMs.

We randomly sample 700 facts across 6 factual relations from the RELATIONS dataset. For each of those examples we block-out information propagation of the *subject*, *non-subject*, and the *prompt-last* token positions for a window of 10 layers around a specific layer $\ell$. The effect of blocking out Conv + SSM information flow for certain layer-token $(\ell - k)$ positions is measured as the relative change in $p(o)$ with $\left( p\left(o \mid a_k^{(\ell)} := \mathbb{E}\left[a^{(\ell)}\right]\right) - p(o) \right) / p(o)$. Figure 7 shows the averaged result and it leads us to draw the following conclusions about how factual information flows in Mamba:

**(a)** The purple lines show that blocking out non-subject information flow in early middle layers can bring down $p(o)$ by up to 50%. Non-subject tokens are used to specify the relation $r$. This observation leads us to believe that Mamba propagates relation specific information to future tokens using Conv+SSM operations in early-middle layers.

**(b)** Interestingly, the green lines (blocking the subject information flow) shows two valleys:

1. The first valley at the early layers is not surprising as Mamba needs to collate information from all the subject tokens in early layers to recognize a subject entity $s$ consisting of multiple tokens.
2. However, the valley at layers 43-48 suggest that Mamba uses Conv + SSM paths in those layers to propagate critical information from the subject to later tokens. This aligns with Figures 5b and 3c, where $s_i$ states in those layers show high indirect effects, indicating their crucial role while recalling a fact.

**(c)** The blue dashed lines indicate the effect of blocking the information of only the subject last token. If the ablation is performed in very early layers, later layers can start to compensate for that. However, the valley around layers 20-21 suggests that Mamba expects to recognize the full subject entity by then in order to recall relevant associations (*enrichment*). Notably, activation patching results for $o_i$ and $s_i$—states that we hypothesize take crucial part in the enrichment process—also show strong indirect effect around that region (Figures 3a, 3b, and 5a). The blue line follows the green line after layer 30. The weaker effect observed might be because ablating subject last token is not always enough to remove all the subject information. For example, in *Eiffel Tower*, *Eiffel* (tokenized as E, iff, el) is more informative than the last token Tower.

These findings align with how factual information flows through attention modules in autoregressive transformer LMs, as observed by Geva et al. (2023) in GPT. However, unlike Geva et al. (2023), we cannot make strong claims about the unique role of the final token position (prompt-last) with this experiment. As we block out information flow to *all* future tokens, the intermediate states in between the ablated $k^{th}$ token and the last token are affected as well.

## 6   Related Works

**Mamba.** Mamba is a recent family of language models that are based on state space models (SSMs). Neural SSM-based models have achieved good performance across different modalities, including vision (Nguyen et al., 2022), audio (Goel et al., 2022), and genomic sequences (Nguyen et al., 2023). Only recently, however, with Mamba, have they become competitive with the language modeling performance of transformers (Gu & Dao, 2023). Like transformers, Mamba contains factual knowledge about real world entities (Grazzi et al., 2024). However, knowledge representation in Mamba (and other LMs based on SSMs) has up to now remained unexplored.

There are few works focused on interpreting Mamba. Ali et al. (2024) identify implicit attention-like matrices formed by Mamba's selective state space layers. Grazzi et al. (2024), while not strictly focused on interpreting Mamba's internals, apply linear probes to Mamba's (decoded) intermediate states during in-context regression tasks. They discover substantial similarities between Mamba and transformer models: both architectures pursue "iterative" strategies, with the task loss falling more or less monotonically as the layer index increases. Finally, in concurrent work, Paulo et al. (2024) apply a different set of interpretability techniques to Mamba, focusing on activation steering and linear probing methods. Like us, they find that approaches developed for transformer models largely transfer to Mamba.

**Locating Factual Knowledge in Language Models.** To make factually correct statements about the world, a LM has to store factual knowledge about real world entities somewhere in its parameters. Understanding how and where a neural network stores knowledge is a core problem for interpretability and it has thus been studied from several perspectives (Ji et al., 2021; Wang et al., 2014). One line of work trains classifiers to probe for properties encoded in model representations (Ettinger et al., 2016; Shi et al., 2016; Hupkes et al., 2018; Conneau et al., 2018; Belinkov et al., 2017; Belinkov & Glass, 2019). However, the flexibility

of these classifiers can lead to overestimating model knowledge and capabilites (Belinkov, 2022). Causal mediation analysis methods (Pearl, 2022) attempt to measure the causal contribution of intermediate states to task performance. Meng et al. (2022a;b) use activation patching to identify key MLP modules for factual recall, highlighting the middle layers at particular token positions as being especially important. Similarly, Geva et al. (2023) apply causal mediation analysis to attention modules, seeking to understand the mechanism of cross-token factual information flow inside transformer LMs.

## 7 Discussion

In this paper we have set out to understand whether the analytical methods and tools developed for transformer LMs can also be applied on the Mamba LMs based on recurrent state-space architecture. Although our experiments have been limited to Mamba-2.8b, the largest available LM of that family, and comparisons to the similarly-sized transformer Pythia-2.8b, the methods we have investigated are general, and can be used to analyze to any state-space model.

Our overall comparisons of Mamba and transformers are positive: with activation patching we have found that, similar to autoregressive transformer LMs, Mamba shows signs of localization at the last subject token and at specific layer ranges while recalling a fact. Although, unlike transformers, Mamba has no MLP modules, we find that their $W_o$ weights can receive rank-one model editing (ROME) edits with good generalization and specificity at a range of layers, similar to $W_{down}$ in Pythia and GPT family of LMs. We have also been able to partially adapt the tools of attention knock-out in Mamba by blocking outgoing information from a token, revealing information flows similar to transformer LMs during factual recall.

The similarity that we have observed between factual recall mechanisms in transformers and Mamba leads us to speculate that the autoregressive language modeling task itself induces a pattern of localized factual recall that is independent of modeling architecture. When constraining a model to process text from beginning to end, the ordering creates a specific bottleneck in the information flows: the end of a subject becomes a singular moment at which recognition of the subject is both possible and useful, and we find that both transformers and Mamba arrange their computations to localize factual recall at that moment. We hypothesize that other future autoregressive LMs architectures should expect to see similar locality in factual recall as well.

In summary, we find that many of the tools used to interpret and edit large transformers can be adapted to work with Mamba, and we are optimistic that those tools will continue to be useful as architectures continue to evolve.

## Ethics

By exploring the factual recall mechanism in Mamba, we potentially improve its transparency, enabling oversight and control. However, the ability to modify facts directly in the model brings with it the potential for abuse, such as adding malicious misinformation or bias.

## Reproducibility

We ran all experiments on workstations with either 80GB NVIDIA A100 GPUs or 48GB A6000 GPUs, using the HuggingFace Transformers library (Wolf et al., 2019) and PyTorch (Paszke et al., 2019). We make use of publicly available datasets COUNTERFACT and RELATIONS in this work.

## Acknowledgements

This research has been supported by a grant from Open Philanthropy (DB, AS), and an NSF Computer and Information Science and Engineering Graduate Fellowship Fellowship (DA). We are also grateful to the Center for AI Safety (CAIS) for sharing their compute resources, which supported many of our experiments. Some of our initial analyses were conducted with a beta version of NNsight (Fiotto-Kaufman et al., 2024) on an implementation of Mamba instrumented for research by Jaden Fiotto-Kaufmann.

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

# A  Datasets

We use two datasets; COUNTERFACT by Meng et al. (2022a) and RELATIONS by Hernandez et al. (2023) in this work.

## A.1  COUNTERFACT

Meng et al. (2022a) developed the COUNTERFACT dataset for evaluating the efficacy of counterfactual edits in language models. It was prepared by adapting PARAREL (Elazar et al. (2021)) and scraping *Wikidata*[3]. The dataset contains $21,919$ requests $\{s, r, o, o^*, \pi^*\}$ where $o$ is the correct answer to the prompt $x = (s, r)$, $o^*$ is the counterfactual edit request, and $\pi^* \sim \mathcal{P}(s, r)$ is a paraphrase of the prompt $x = (s, r)$ to test for generalizability (PS). Each of the records also contain some neighborhood prompts $\pi_N$ to test for specificity (NS) and some generation prompts $\pi_G$ to test if LM generation post-edit is fluent and consistent with the edit. Please refer to Meng et al. (2022a) for details on the curation of this dataset.

We evaluate ROME performance in Mamba-2.8b (Figure 6a) and Pythia-2.8b (Figure 6b) on the first 2000 records from COUNTERFACT.

## A.2  RELATIONS

The RELATIONS dataset introduced in Hernandez et al. (2023) consists of 47 relations of 4 types: *factual*, *linguistic*, *bias*, and *commonsense*. A relation $r$ is an association between two entities. For example, the relation, $r = $ *professionally played the sport* connects the subject $s = $ *Michael Jordan* with the object $o = $ *basketball*. The dataset contains a set of $(s, o)$ for each relation $r$.

In the scope of this paper, we only utilize the 26 *factual* relations from this dataset. We evaluate LRE in Mamba and Pythia for all the 26 factual relations. We also use this dataset for locating key fact-mediating states in Section 3 and Appendix D. We randomly sample 400 examples $(s, r, o)$ across 6 different factual relations - *place in city*, *country capital city*, *person occupation*, *plays pro sport*, *company hq*, and *product by company*. For each of these examples we randomly select another example within the same relation $(s^*, r, o^*)$ such that $s \neq s^*$ and $o \neq o^*$. The average indirect effect (IE) of applying activation patching over these 400 examples is depicted on Figures 2b, 3, 5 for Mamba-2.8b) and on Figure 9 (for Pythia-2.8b). We use the same set of 6 relations in Section 5 where we adapt attention knock-out to Mamba.

# B  Challenges in Performing Attention Knock-out in Mamba

Attention heads in autoregressive transformer LMs and Conv + SSM operations in Mamba play a similar role: bringing/retaining information from the past tokens. Attention *"knock-out"* is a type of causal mediation analysis that tries to understand information flow in transformer LMs by cutting off information propagation from $k^{th}$ token to $q^{th}$ token position. In transformers, each of the attention heads in an attention module $attn^{(\ell)}$ calculates an *attention matrix* L, where $L_{q,k}$ quantifies how much attention is being paid to the $k^{th}$ token by the $q^{th}$ token with this specific attention head (see Vaswani et al. (2017) for details on the attention operation). We can block the information flow from $k^{th}$ token to $q^{th}$ token via a specific attention head by simply setting $L_{q,k} := -\infty$ in the forward pass.

For Mamba, Ali et al. (2024) show that the amount of information retained in the $q^{th}$ token state $s_q^{(\ell)}$, from the *convolved* state at $k^{th}$ token $c_k^{(\ell)}$ (where $k < q$), after the selective-SSM operation (see Equations 4 and 5) can be visualized as an *attention matrix* per channel. Since the selective-SSM operation is linear, the information retained in $s_q^{(\ell)}$ from $c_k^{(\ell)}$ can

---

[3]www.wikidata.org/wiki/Wikidata:Main_Page

be calculated accurately as $\tilde{\alpha}_{q,k}^{(\ell)} = \overline{C}_q^{(\ell)} \left( \prod_{i=k+1}^{q} \overline{A}_i^{(\ell)} \right) \overline{B}_k^{(\ell)} c_k^{(\ell)}$, where $\overline{A}_i^{(\ell)}$, $\overline{B}_i^{(\ell)}$, and $\overline{C}_i^{(\ell)}$ are input-dependent parameters for the $i^{th}$ token. See Gu & Dao (2023) and Ali et al. (2024) for details on selective-SSM operation. We ask: can we block the information flow from the $k^{th}$ token to the $q^{th}$ token in Mamba by subtracting out $\tilde{\alpha}_{q,k}^{(\ell)}$ from $s_q^{(\ell)}$? If so, attention knockout experiments in Mamba become feasible.

We find that blocking information flow via Conv + SSM operation through this specific edge from the $k^{th}$ token to the $q^{th}$ token can be challenging in Mamba. Note that, since $c_k^{(\ell)}$ is a *convolved* state with a receptive field of size 4 in Mamba-2.8b, the states $c_{k+1}^{(\ell)}$, $c_{k+2}^{(\ell)}$, and $c_{k+3}^{(\ell)}$ also retain information from $a_k^{(\ell)}$. Which means that even if we subtract $\tilde{\alpha}_{q,k}^{(\ell)}$ from $s_q^{(\ell)}$, these states can *"leak"* information about $a_k^{(\ell)}$ to $s_q^{(\ell)}$. To stop this leakage, we would want to subtract from $s_q^{(\ell)}$ all the information retained from $a_k^{(\ell)}$ via $c_{k+1}^{(\ell)}$, $c_{k+2}^{(\ell)}$, and $c_{k+3}^{(\ell)}$ states as well. However, accurately calculating this is challenging because of the SiLU non-linearity after Conv1D (see Equation (4)).

In our initial experiments we tested subtracting only $\tilde{\alpha}_{q,k}^{(\ell)}$ from $s_q^{(\ell)}$. But we found that Mamba-2.8b could often refer to the $k^{th}$ token from the $q^{th}$ token in copy and factual recall tasks.

## C  Isolating The Contribution of $W_o^{(\ell)}$

Recall from Figure 1 and Equation (2) that when $o_i^{(\ell)}$ is restored, the $s_i^{(\ell)}$ and $g_i^{(\ell)}$ are restored as well. To isolate the contribution of only $W_o^{(\ell)}$ we subtract out $\mathrm{IE}_{s_i^{(\ell)}} + \mathrm{IE}_{g_i^{(\ell)}}$ from $\mathrm{IE}_{o_i^{(\ell)}}$ and plot the results on Figure 8. Notice that subtracting $\mathrm{IE}_{s_i^{(\ell)}}$ cancels out the high indirect effect at the late site shown by later layers at the last token position. But, together $\mathrm{IE}_{s_i^{(\ell)}} + \mathrm{IE}_{g_i^{(\ell)}}$ cannot cancel out high $\mathrm{IE}_{o_i^{(\ell)}}$ observed at the early site, that is early-mid layers at the last subject token. This reconfirms the mediating role of $W_o^{(\ell)}$ at the early site while recalling a fact.

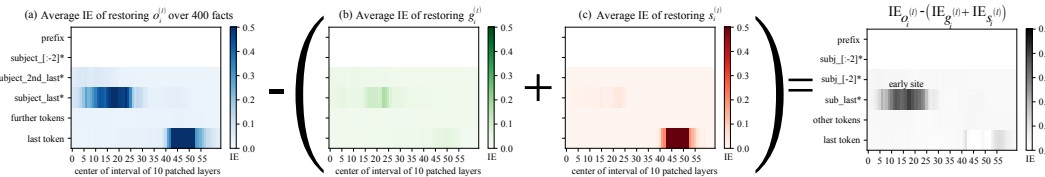

Figure 8: Isolating the contribution of $W_o^{(\ell)}$. $\mathrm{IE}_{s_i^{(\ell)}} + \mathrm{IE}_{g_i^{(\ell)}}$ subtracted from $\mathrm{IE}_{o_i^{(\ell)}}$. Notice that $\mathrm{IE}_{o_i^{(\ell)}} - \left( \mathrm{IE}_{s_i^{(\ell)}} + \mathrm{IE}_{g_i^{(\ell)}} \right)$ still shows higher causal effect at the early site (more pronounced than $\mathrm{IE}_{g_i^{(\ell)}}$) while the high causal effect at the late site cancels out.

# D  Locating Key Modules in Pythia-2.8b

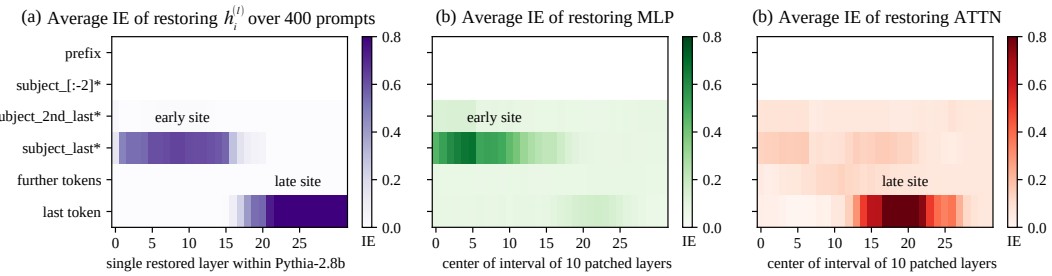

Figure 9: Average indirect effect of residual state, MLP, and attention outputs in Pythia-2.8b over 400 facts. For MLP and attention outputs a window of 10 layers around $\ell$ is restored, as restoring just one layer barely shows visible patterns.

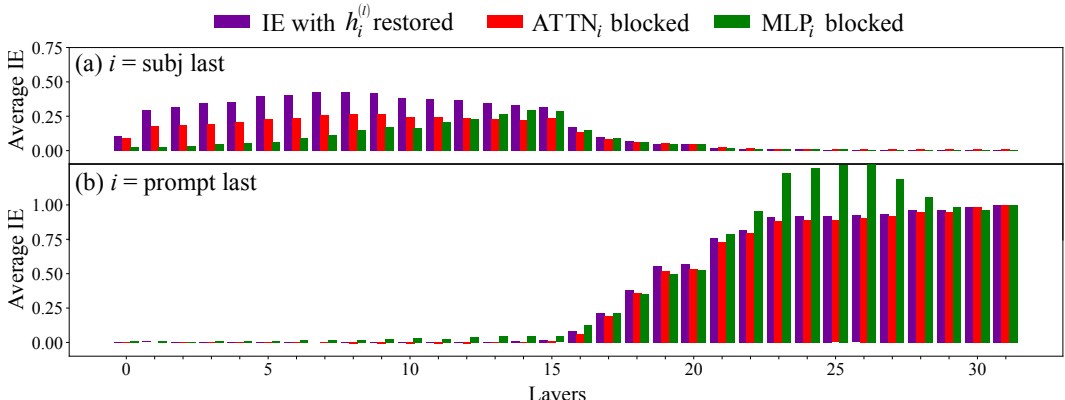

Figure 10: Impact of ablating $\text{ATTN}_i$ or $\text{MLP}_i$ on $\text{IE}_{h_i^{(\ell)}}$ for **(a)** *subject last* and **(b)** *prompt last* token positions on Pythia-2.8b

# E  LINEARITY OF RELATION EMBEDDING (LRE)

With activation patching we can identify *where* facts are located inside a LM. We are also interested in understanding *how* LMs extract this information given $x = (s, r)$. Figures 2b and 5 show a clear separation of roles in early-mid and later layers in Mamba. We observe a similar phenomenon in autoregressive transformer LMs (Meng et al., 2022a;b; Geva et al., 2023). According to Geva et al. (2023), in transformer LMs, the subject entity representation **s**, at the subject last token position, goes through an *enrichment* process, mediated by the MLP in the early-mid layers, where **s** is populated with different facts/attributes relevant to the subject entity $s$. Then, at the last token position, attention modules perform a *query* on the *enriched* **s** to extract the answer to the prompt $x = (s, r)$. Hernandez et al. (2023) approximate the *query* operation performed on the *enriched* **s** for a specific relation $r$ by taking the first order Taylor series approximation (LRE) of the LM computation $F$ as

$$F(\mathbf{s}, r) \approx \beta J_\rho \mathbf{s} + b$$

$$\text{where } J = \mathbb{E}_{\mathbf{s}_i, r} \left[ \left. \frac{\partial F}{\partial \mathbf{s}} \right|_{(\mathbf{s}_i, r)} \right] , \quad b = \mathbb{E}_{\mathbf{s}_i, r} \left[ F(\mathbf{s}, r) - \left. \frac{\partial F}{\partial \mathbf{s}} \, \mathbf{s} \right|_{(\mathbf{s}_i, r)} \right] , \tag{8}$$

$$\beta \text{ is a scalar }, \text{ and } \quad \rho \text{ is the rank of } J$$

Hernandez et al. (2023) show that for a range of different relations it is possible to achieve a LRE that is faithful to the model computation $F$ by averaging the approximations of $J$ and $b$ calculated on just $n = 5$ examples. We utilize LRE to understand the complexity of decoding

*factual* relations in Mamba. We find the hyperparameters $\beta$, $\rho$ and the layer $\ell$ (where to extract the enriched **s** from) using grid search. For mathematical and implementation details, see Hernandez et al. (2023).

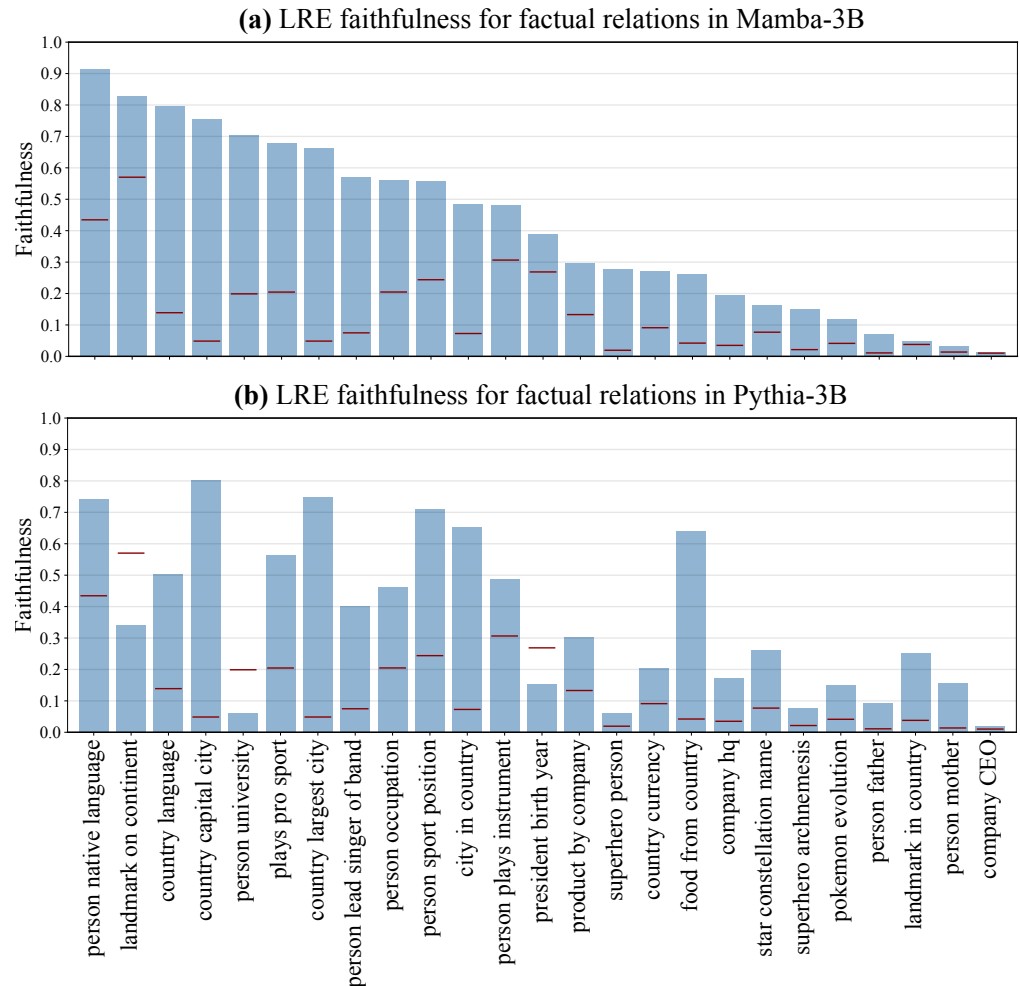

Figure 11: Relation-wise LRE *faithfulness* to the LM decoding function $F$ for **(a)** Mamba-2.8b and **(b)** Pythia-2.8b. The relations are sorted according to their LRE faithfulness in Mamba-2.8b. Horizontal red lines per relation indicate random-choice baseline. We only present results for the *factual* relations in the RELATIONS dataset.

We plot the *faithfulness* of LRE with $n = 5$ samples on Figure 11 for Mamba-2.8b and similar sized Pythia. The metric *faithfulness* represents the portion of facts $(s, r, o)$ that can be correctly retrieved if the LM computation $F(\mathbf{s}, r)$ is replaced with LRE($\mathbf{s}$), a simple affine transformation.

We only calculate LRE for the *factual* relations in the RELATIONS dataset. Figure 11a shows that only for 10 out of 26 factual relations can a *linear* LRE achieve more than 50% *faithfulness*. For comparison, in the same sized Pythia-2.8b LRE achives $> 50\%$ *faithfulness* for 11 factual relations (see Figure 11b). And, in both Mamba and Pythia, LRE fails to achieve good *faithfulness* for the relations where the *range* (the number of unique answers) is large. These findings align with what Hernandez et al. (2023) observed on GPT and LLaMA models; suggesting that, similar to transformer LMs, factual knowledge might be heterogeneously represented for different relations in Mamba.

## E.1 LRE Performance Across Different Layers

Besides *faithfulness* Hernandez et al. (2023) introduced another metric *causality* to measure the performance of LRE. Since LRE is a linear function, it is invertible. Assume that for a fact $(s, r, o)$ LRE can faithfully replace LM computation $F(\mathbf{s}, r)$. Then given the representation $\mathbf{o}^*$ of another object $o$, $\mathbf{J}^{-1}(\mathbf{o}^* - \mathbf{o})$ should give us a $\Delta\mathbf{s}$, such that when added to $\mathbf{s}$, $\tilde{\mathbf{s}} := \mathbf{s} + \Delta\mathbf{s}$, the model computation $F(\tilde{\mathbf{s}}, r)$ should generate $o^*$. See Hernandez et al. (2023) for details on this.

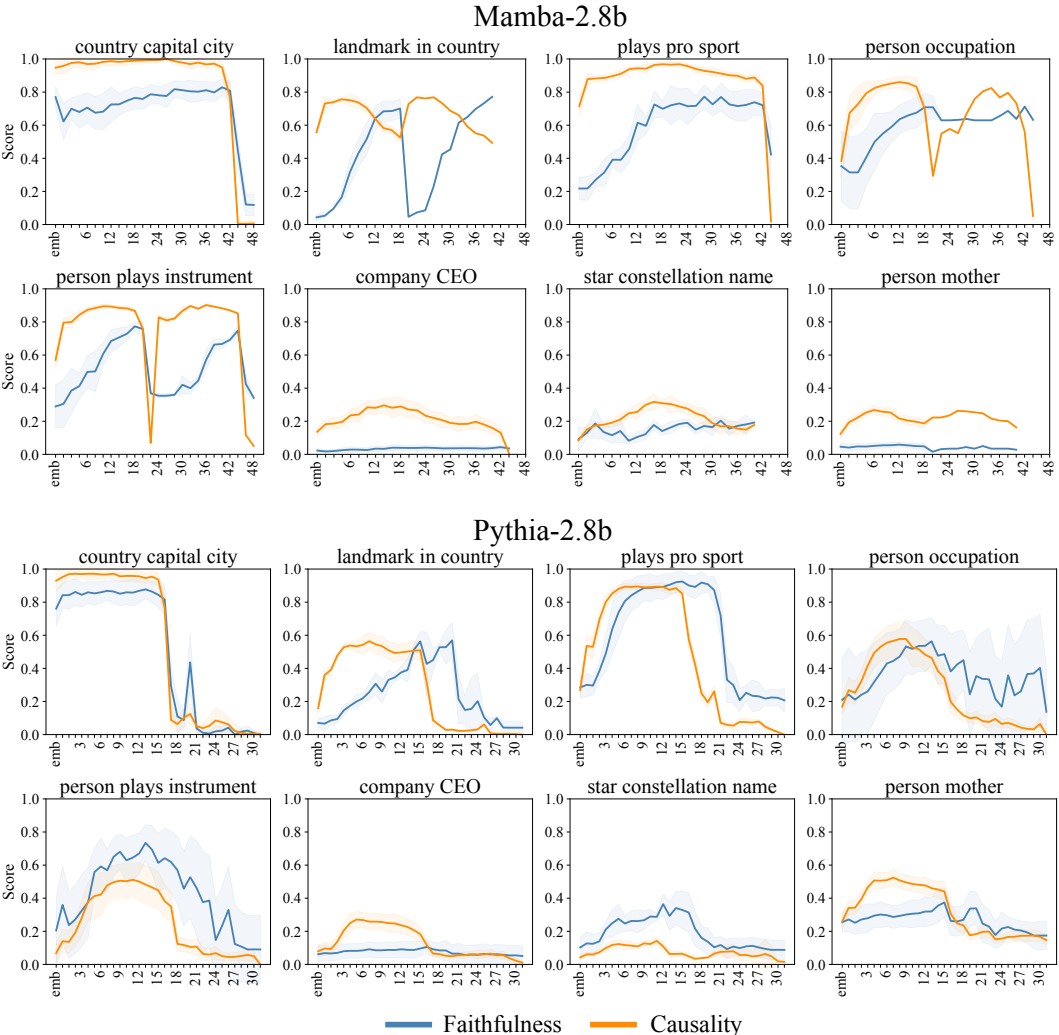

Figure 12: For Mamba, we only perform sweep till layer 48, as Figure 5 suggests negligible activity for later layers at the subject last token

## F Activation Patching results on Mamba-2.8b

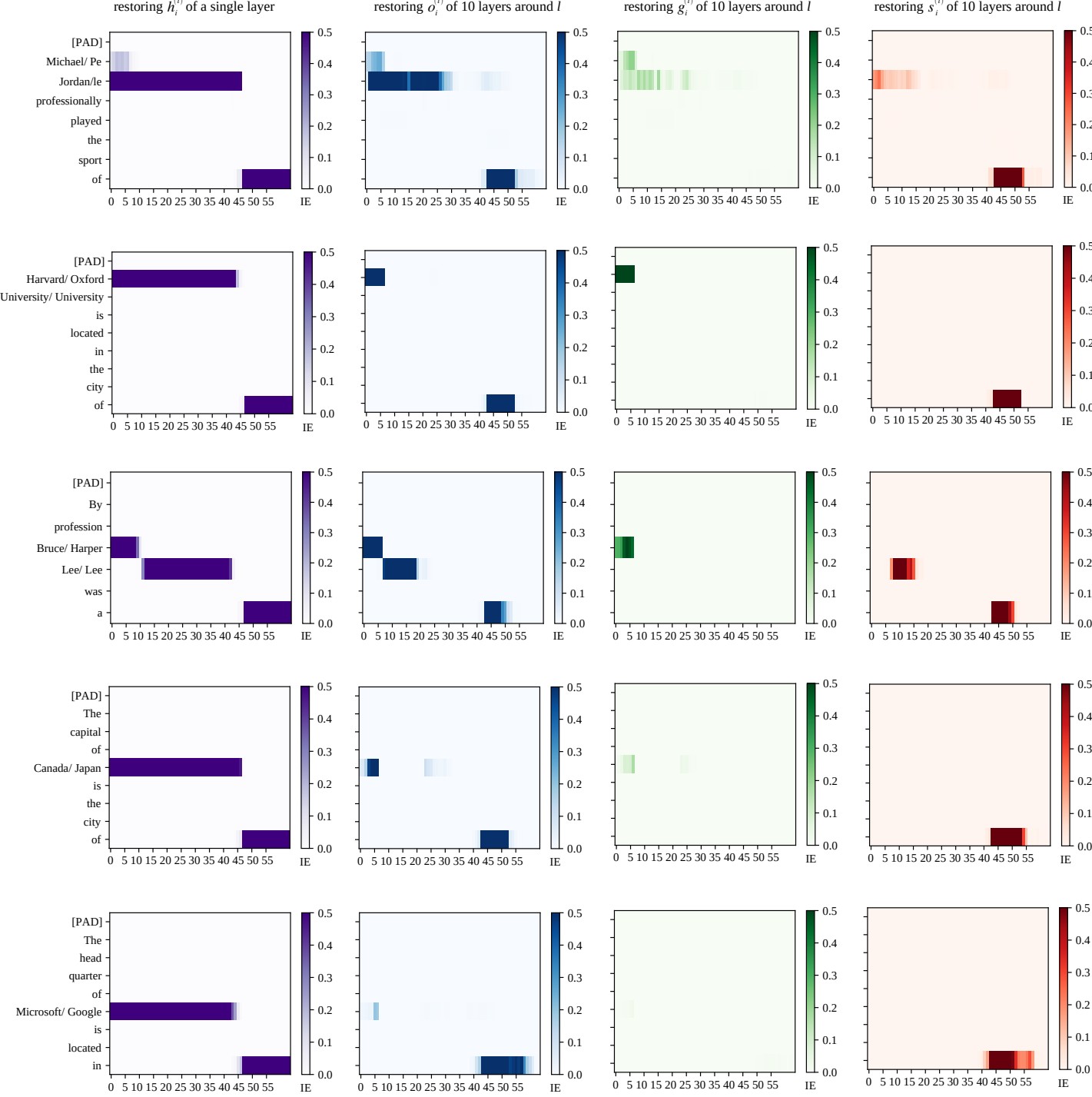

