# OpenReview forum: "Locating and Editing Factual Associations in Mamba"
_colmweb.org/COLM/2024/Conference — COLM_

### Official Review · Reviewer_rwMm · 2024-05-11

**Rating:** 7
**Confidence:** 4
**Ethics Flag:** 1

**Summary:**

This paper presents a thorough investigation of factual knowledge in Mamba. The authors adapt and apply several techniques used in Transformer LMs to investigate where and how Mamba encodes facts. Specifically, the authors apply (i) activation patching to locate facts, (ii) ROME to edit facts, and (iii) LRE to understand how Mamba extracts factual knowledge. In doing this, the authors find many similarities between Transformer LMs and Mamba for factual knowledge, as well as some differences and challenges, such as blocking the information from between two tokens due to the receptive field of the convolutional layer.

**Questions To Authors:**

1. Which dataset did you use in Sections 3 and 6?

**Reasons To Accept:**

1.  A thorough investigation of factual knowledge in Mamba models.
2.  The paper provides insights across different axes, from where facts are stored, to how they can be encoded, as well as how to edit them.
3. The paper is generally well written and organized.

**Reasons To Reject:**

I do not have any strong reasons to reject the paper. That said, some parts could be improved:
1. I found it hard to follow the experiments on “path-specific effects” (page 5). This subsection could be improved perhaps by bringing Figure 5 closer to the text and making more explicit connections so the reader can better visualize the process.
2. Present a succinct table or text that compares Mamba with Transformer LMs so that the reader can quickly glance at your key results before delving into the corresponding sections.

---

> ### Author Rebuttal · Authors · 2024-05-29
>
> Thank you for the positive review and your suggestions on how to improve our paper. We will try our best to incorporate your feedback in the next revision.
>
> > "I found it hard to follow the experiments on “path-specific effects” (page 5). This subsection could be improved ... "
>
> We will update parts of section 3 to make out path-specific experiment setup easier to understand. Thank you for your suggestions.
>
>
> > "Which dataset did you use in Sections 3 and 6?"
>
> Thanks for bringing this to our attention! We had to move the dataset section in Appendix A due to the page limit on the main paper. The activation patching results (Figures 2b, 3, and 5 in section 3; also Figs 8, 9 and 10 in the appendix) are averaged over 400 facts sampled from 6 different relations in the Relations dataset ([Hernandez et al, 2023](https://arxiv.org/pdf/2308.09124)) - *place in city*, *country capital city*, *person occupation*, *plays pro sport*, *company hq*, and *product by company*.
>
> The experiment in Section 6 (continued on Appendix D) was conducted on 700 examples randomly sampled from the same set of relations in the Relations dataset. We will update the text in relevant sections and the figure captions to clarify this.

---

> > ### Comment · Reviewer_rwMm · 2024-06-06
> > **Acknowledgement of rebuttal**
> >
> > Dear authors, thank you for clarifying my questions. I hope you will add the missing details in the extra page provided for accepted papers.

---

> ### Author Response · Authors · 2024-06-02
>
> Dear Reviewer rwMm,
>
> Thank you for your suggestions on how to improve our paper. As we have a short time left in the discussion period, a gentle nudge to please let us know if you are satisfied with our responses. Kindly let us know if you have any further feedback.
>
> Thank you!

---

### Official Review · Reviewer_SQMx · 2024-05-11

**Rating:** 5
**Confidence:** 3
**Ethics Flag:** 1

**Summary:**

This work tries to analyze the mechanism of factual recall in Mamba following the pattern that is previously employed to analyze Transformer-based language models. Specifically, this work reveals that the activation patching technique can be used to specify model components that relates to outputs the correct objects in factual recall. However, applying rank one model editing (ROME), linearity of relation embedding, and knocking out attention in Mamba demonstrate different behavior.

**Questions To Authors:**

- The term 'early sites' and 'late sites' lack a clear definition throughout the work.
- The claim 'transformer MLP outputs have effects in early site and not the late site' needs reference or connection to other parts to be better understood.

**Reasons To Accept:**

This work adds value in aligning the analytical tools in Transformer-based architectures to non-transformer architectures like Mamba.

**Reasons To Reject:**

I am a bit confused about what narrative this work tries to build: the previous framework in analyzing how factual recall happens in transformer-based language models partially works in Mamba, or how factual recall happens in Mamba based on some existing works that are done in transformer-based language models? So far it reads like the former, which makes different parts not connected strongly under the architecture of Mamba.

---

> ### Author Rebuttal · Authors · 2024-05-29
>
> Thank you for your constructive review. We are glad to hear that you found our work valuable.
>
>
> > "I am a bit confused about what narrative this work tries to build ... "
>
> We are actually interested in a broader question confronting the field of interpretability - *to what extent do the analytical tools and our insights on certain mechanisms developed for one architecture (such as Transformers) generalize to a very different architecture?* This work is a case study, exploring this question by trying to understand the factual recall mechanism in Mamba with methods that have been effective in understanding how Transformer LMs do so. Our investigation reveals that these methods can often be applied directly or with some adaptations to account for architectural differences. Additionally, we find that, despite significant architectural differences, Mamba shares many high-level similarities with transformer LMs when it comes to factual recall.
>
> We will update the abstract and relevant sections in the introduction to clarify our motivation and the narrative of this paper.
>
> > "The term 'early sites' and 'late sites' lack a clear definition throughout the work."
>
> We will annotate Figures 2, 3, and 9 in the revised draft to clarify what we refer to as “early site” and “late site”. Thanks for bringing this to our attention.
>
>
> > "The claim 'transformer MLP outputs have effects in early site and not the late site' needs reference or connection to other parts to be better understood."
>
> Thank you for catching this. This claim was made comparing Figure 3 (Mamba 2.8b) in Section 3 with Figure 9 (Pythia 2-8b) in the Appendix. Causal mediation analysis (Figure 9) shows that, when Transformer LMs recall a fact, MLP contributions are pronounced at the early site (subject last token - early to mid layers), while the high causal effect at the late site (prompt last token - later layers) is due to the Attention mechanism. We will revise the text to clarify this.
>
>
> Let us know if you have further concerns or feedback to improve our paper. And kindly consider adjusting your score upwards if you are satisfied with our response.

---

> > ### Comment · Reviewer_SQMx · 2024-06-06
> >
> > Thanks for the reply from the authors and more information about the questions. Since the theme of the work is to investigate how much the current analysis tools can be extended to non-Transformer architecture and the results shows a partial matching of the transfer, the work is better to include more high-level guideline about the failure part and how many properties are architecture-agnostic (as well as agnostic to scaling model parameters) a which can be considered for future novel architectures. So far, the work does not seem to bring a sufficient answer to this question. I would rather remain my score unchanged.

---

> ### Author Response · Authors · 2024-06-02
>
> Dear Reviewer SQMx,
>
> We have a short time left in the discussion period, so a gentle nudge to please let us know if you are satisfied with our responses. As a reminder, we plan to revise the paper in following ways to incorporate your feedback:
>
> 1. Clarify the narrative and motivation in the Abstract And Introduction.
> 2. "early site" refers to the prominent indirect effects noticed at the subject last token in early to mid layers. And "late site" refers to the high causal effect at the prompt last token in later layers. We plan to annotate Figures 2, 3, and 9; also revise the text in relevant sections to clarify this.
>
> We invite you to share any additional feedback and help us improve our paper. And kindly consider increasing your score if your concerns have been addressed.
>
> Thank you!

---

### Official Review · Reviewer_o7rW · 2024-05-13

**Rating:** 7
**Confidence:** 3
**Ethics Flag:** 1

**Summary:**

The work adapts a series of information tracing and editing techniques originally developed for transformers to the Mambda state-space model. The general finding is that facts are stored (and can be edited) in a similar way in Mamba as they are in transformers (the representatitve transformer model here beign Pythia).

This is a nice exercise in transfering knowledge across model architectures. The authors have considered a wide range of information tracing methods and a well-established editing method - ROME. The work seems solid and could become quite impactful if state-space models take off.

**Questions To Authors:**

Typos:
- "In this we are able"
- "the evolution of hidden state"
- Figure 5 is missing an x-axis label

**Reasons To Accept:**

Clear motivation and technical contribution.

**Reasons To Reject:**

The authors should include a real discussion (rather than what is now called Discussion but is really a Conclusion) and address the question: how would the approaches proposed here generalize to other state-space models? I.e. how much can a state-space model deviate from the exact Mamba architecture and still be analyzed or manipulated with the methods presented here?

---

> ### Author Rebuttal · Authors · 2024-05-29
>
> Thank you for your positive review. We are glad that you found our work well motivated.
>
> > "... how would the approaches proposed here generalize to other state-space models?"
>
> In this paper we investigate two primary questions -
>
> 1.  Can the analytical tools and methods developed to understand factual recall mechanisms in transformer language models be applied across different language modeling architectures?
> 2. To what extent do our insights on factual recall mechanisms in transformer LMs (at a high level) generalize across different types of LMs?
>
> We limit our experiments on Mamba-2.8b, the best performing language model among all the linear-time RNN approaches; including SSMs, RWKV, and other architectures ([Gu and Dao, 2023](https://arxiv.org/abs/2312.00752)). Our experiments yield overall positive results on both questions. This gives us reason to believe that the approaches and findings in this paper are not specific to any language modeling architecture, but will (to some extent) generalize to a range of other SSM architectures; the main assumption is that it is an **autoregressive** language model.
>
>
> As we discussed in Section 6 and Appendix D, replicating path-dependent attention knock-out experiments in Mamba pose specific challenges due to the use of convolutions and a non-linearity  in conjunction with pure SSM state. Other SSM architectures without this complexity may allow similar experiments as well.
>
>
> Thank you for your suggestion. We will clarify our stance in the Discussion section in the next revision.
>
>
> > Typos
>
> Thanks for the deep read and catching the typos. We will address them in the revised draft.

---

> ### Author Response · Authors · 2024-06-02
>
> Dear Reviewer o7rW,
>
> Thank you again for your suggestions on how to improve the Discussion section in our paper and for catching the typos. We have a short time left in the discussion period, so a gentle nudge to please let us know if we are satisfied with our responses. Kindly let us know if you have any additional feedback.
>
> Thank you!

---

### Official Review · Reviewer_kU61 · 2024-05-16

**Rating:** 7
**Confidence:** 3
**Ethics Flag:** 1

**Summary:**

**Paper summary**

The paper studies mechanistic interpretability of factual knowledge in the context of a pretrained Mamba language model. It borrows heavily from recent techniques that have been explored in the context of Transformers,  applying or adapting interpretability techniques specifically for the Mamba model. Using a few different probes, the paper provides some analysis comparing specificity of factual knowledge location in Mamba with Transformers as well as highlighting some limitations in reusing existing techniques.

**Review Summary**

Overall, I found this to be a useful study that has some flavor of a replication study (reapplying existing techniques and analysis) to a slightly new model architecture (Mamba vs Transformer). I will recommend acceptance of the paper as the work is clearly written and presented and thorough application of interpretability techniques to the Mamba architecture. The main drawback of the paper is that some of the results are of limited use in their current form. Aside from general insights on layer/neuron representation of entities, it is still unclear how one could make use of this analysis in practice.

**Questions To Authors:**

- I could not find if there is a specific dataset for Section 3. It mentions that there are “400 facts” analyzed, but I could not find where these facts are derived from.

**Reasons To Accept:**

- The paper is well written and clearly presented. The background on Mamba and the studied techniques are useful.

- The investigation of interpretability methods to novel LM architectures is a useful type of study reproduction. As a field, we should invest in novel architectures and make sure our techniques can be robust to architectural changes.

- The adaptation of the activation patching to Mamba, allowing several techniques to be adapted from their Transformer counterparts, is well motivated and executed.

- The analysis of the results, while not unexpected following prior results on Transformers, is clear and uses sound experiments.

**Reasons To Reject:**

- While some of the analysis presented is interesting, it is unclear how to make use of some of the observations in practice. One example is the results from the LRE (Linearity of relation embedding) section. The faithfulness scores of the LRE are pretty low making it hard to claim that these representations truly capture internal representation of what goes inside the neural network.

- Similarly for the attention knock-out investigation. Suppose we can more clearly understand information flow inside of Mamba (or Transformer); what are practical applications of this knowledge? Do we gain better insights into training new models? Fixing factuality errors? Better understand model confidence about its own knowledge?

---

> ### Author Rebuttal · Authors · 2024-05-29
>
> Thank you for your positive review. We are pleased to know that you found our work well-motivated and useful.
>
> > “ … What are practical applications of this knowledge”?
>
> Although not asked as a question, reviewer `kU61` was skeptical how or if our investigation on LRE (Section 5) and factual information flow (Section 6) have any practical applications. We believe this is an important question and here is our position on this -
>
> All of these methods, including understanding information flow and LRE, have potential applications in helping debug model behavior. Understanding the role of different components inside the model while performing a task may reveal potential “bugs” in the mechanism; like: Is the mechanism missing something crucial? Or is the model doing something unnecessary/weird and why? …
>
> As for LRE, [Hernandez et al, 2023](https://arxiv.org/pdf/2308.09124) present a tool called *attribute lens* based on it, which can be directly applied to the latent of different token-layer positions in a free-form text to visualize the model’s current belief. If a model utters an incorrect fact for which the relation is *linear*, *attribute lens* could be used to visualize whether information retrieved early in the model is incorrect (suggesting the LM might not have learned the association from training data), or if the mistake is happening later (suggesting the model is making a mistake due to something else in the context). However, *attribute lens* will fail to elicit information faithfully for non-linear relations.
>
> We recognize that full end-to-end debugging will eventually require a better understanding of processing of information at every stage, not just factual association. But partial insights are already helpful in building intuition and they may help direct future research.
>
>
> > "I could not find if there is a specific dataset for Section 3 ..."
>
> Thanks for bringing this to our attention. Due to space limitations in the main paper we had to move the dataset section in the appendix (Appendix A). The "400 facts" mentioned in the results of activation patching experiments (Figures 2b, 3, and 5 in Section 3) were sampled from 6 different relations in the Relations dataset (Hernandez et al, 2023) - *place in city*, *country capital city*, *person occupation*, *plays pro sport*, *company hq*, and *product by company*. We will update the figure captions and text in relevant sections to refer to Appendix A in the next revision to clarify this.

---

> > ### Author Response · Authors · 2024-06-02
> >
> > Dear Reviewer kU61,
> >
> > We have a short time left in the discussion period, so a gentle nudge to please let us know if we are satisfied with our responses to your concerns. Kindly let us know if you have any further feedback or suggestions to improve our paper.
> >
> > Thank you!

---

### Author Response · Authors · 2024-06-05

We are grateful to our reviewers for their thoughtful and constructive feedback. We plan to revise our paper to incorporate their suggestions, which we believe will strengthen it. Here is a summary of changes we intend to make:

* [`kU61`, `rwMm`] Revise the text and captions of figures in Sections 3 and 6 to clarify the dataset used to perform those analyses.
* [`o7rW`] In the Discussion section, include our opinion on to what extent the approaches explored in this paper are generalizable across different SSM architectures.
* [`kU61`, `o7rW`] Revise the text in the Abstract, Introduction, and relevant sections to clarify the narrative and motivation of our work.
* [`SQMx`] Update the text and annotate Figures 2, 3, and 9 in our paper to clarify “early site” and “late site”.
* [`rwMm`] Revise the description of path-dependent experiment setup in Section 3 to make it easier to understand.


We have responded to each reviewers questions/concerns in separate responses. We invite the reviewers to share additional comments, questions, or suggestions to improve our paper. And kindly consider adjusting scores upwards if your concerns have been addressed.


Thanks!

---

### Decision · Program_Chairs · 2024-07-10

**Decision:**

Accept

**Comment:**

The submission applies several recent mechanistic interpretability techniques to Mamba with the key question being *To what extent do those technique generalize between different architectures?*. The reviewers agree that while the submission is not introducing a new method, it is a useful and timely study in interpretability of SSMs. It is recommended that the authors rework the discussion section so that it becomes less of a summary but spends more time on what classes of interpretability and editing methods seem most promising for Mamba and related models.

The authors promise to open-source the code which I believe would be quite useful for other researchers.